# Spatiotemporal Characteristics of Food Supply–Demand Balance in Uzbekistan under Different Scenarios

**DOI:** 10.3390/foods12102065

**Published:** 2023-05-20

**Authors:** Xinzhe Song, Yanzhao Yang, Chiwei Xiao, Chao Zhang, Ying Liu, Yuanqing Wang

**Affiliations:** 1Institute of Geographic Sciences and Natural Resources Research, CAS, Beijing 100101, China; songxz.21b@igsnrr.ac.cn (X.S.);; 2College of Resources and Environment, University of Chinese Academy of Sciences, Beijing 100049, China; 3Key Laboratory of Carrying Capacity Assessment for Resource and Environment, Ministry of Natural Resources, Beijing 101149, China; 4Faculty of Geography, Yunnan Normal University, Kunming 650500, China

**Keywords:** supply–demand balance, food production and consumption, land carrying capacity, dietary nutrition, Uzbekistan

## Abstract

The food supply–demand balance is a perpetual concern for many countries, especially developing countries, such as Uzbekistan. Using the land resource carrying capacity model, here, food supply and demand for the cereals and calories in Uzbekistan during 1995–2020 were revealed. Despite increased demand for cereals and calories, unstable crop production has led to volatile growth patterns. The carrying capacity of cropland resources under Uzbekistan’s consumption standard shifted from overload to surplus and then to balance. Moreover, the carrying capacity of cropland resources under the healthy diet standard moved from balance to surplus in the past 25-years. Additionally, the calorific equivalent land resource carrying capacity under Uzbekistan’s consumption standard fluctuated, with the carrying state shifting from balance to surplus, and the healthy diet standard still in overload. These findings can help guide sustainable production and consumption strategies in Uzbekistan and other countries by analyzing the consumption structure and changes in supply and demand relationships.

## 1. Introduction

With continued global population increase, consumption growth, climate change, and resource crises, meeting the demand for food has become a formidable challenge [1]. Some global initiatives (e.g., the 2030 United Nations Sustainable Development Goals) [1,2] and challenging scientific problems (i.e., how many people can the Earth sustain?) [1] have repeatedly emphasized the significance of food security and propelled food supply and demand to the forefront of the debate on sustainable development [3]. The sustainability of food production and consumption is fundamental to human survival and essential to the development of some countries and regions, especially developing countries. The balance of the relationship between food production and consumption reflects not only the level of food production but also the quality of local diets [3]. Therefore, characterizing the patterns and trends of the food supply–demand balance is of great significance for local sustainable development.

Maintaining and improving food security is one of the major challenges facing the world in the 21st century [1]. Increasingly frequent and severe extreme weather events and public health emergencies (e.g., COVID-19) are impacting supply chains, along with rising consumer food prices and an increase in the number of people around the world who cannot afford healthy diets, adding to the challenges of daily life [4,5]. In particular, given the impact of pandemics on the food and agricultural sectors, sustainable food production are more important than ever. Currently, many scholars and institutions have recognized that the principles of controlling food production and consumption are necessary for countries to develop appropriate agricultural strategies and land-use designs, and to provide scientific references and guidance for local governments [6,7]. Given the impact of pandemics (e.g., COVID-19) on the food and agricultural sectors, a sustainable food supply and demand balance is more important than ever [4]. 

As an invaluable measure of the balance between cereal and food supply and demand in a region, land resource carrying capacity (LCC) is crucial to the stability of food production and consumption in a country or region [8]. The term of LCC was first introduced by Park in 1921 and mainly focuses on the number of people that can be supported by land resources [9]. To date, LCC research has gradually expanded from cereal supply–demand balance analysis to food supply–demand balance research. Cereal and nutrition concepts have been gradually adopted in studies related to whether food production can meet the demands of the existing population, whether production and consumption structures are sustainable and whether the supply–demand balance can be achieved [10,11,12]. On the one hand, research on food production focuses on healthy diet [13], animal-based foods [14], food safety and sustainability [15]. On the other hand, some studies focus on the food consumption-influencing factors and structure, as well as the nutritional intake structure [8,16]. Among them, mathematical statistics [17], the grain equivalent method [18], ecological footprint [19], life cycle assessment [20], the energy method and linear analysis [21] are widely used to delineate food production and consumption. With the development of Geographic Information Systems (GIS), spatial mapping methods are being integrated into research methods [22]. However, the study of food production and consumption still lacks a suitable and unique computational standard for assessing different types of food products. This hinders a thorough understanding of local food production and consumption, especially in developing countries such as Uzbekistan. The LCC model can better link food production and consumption to explore the supply and demand balance of national food consumption and ensure national food security.

To date, research on food production in Uzbekistan has focused on yield analysis, the relationship with health, crop diversity, spatial patterns and agronomic practices and their influencing factors [23,24,25,26,27]. Food consumption studies have mostly concentrated on food waste, eating habits and health and the factors influencing consumption [23,28,29,30]. In recent years, many scholars have conducted in-depth studies on the assessment and forecasting of food production, impact of food trade, and food security [26,30]. In Uzbekistan, some academics have focused on wheat supply and demand forecasts [24,25], the mechanisms of influence between land policy reform [31], climate change [32] and food supply [33], the impact of food shortages and agricultural development patterns [27]. Additionally, many scholars are concerned about nutritional security [31]. For example, using small-scale survey data, they focus on the nutritional status of children and women and its relationship to food security, dietary diversity and the influencing factors [33]. Overall, previous studies mainly focused on food security and nutrition security, and seldom considered national macroscopic cereal security. Thus, there is a scarcity of research combining cereal production, consumption, and the supply–demand relationship. In particular, research on food in Uzbekistan mostly focuses on food consumption or improving the quantity of food [24,34]. However, comprehensive studies linking both production and consumption patterns and their changes, the food supply–demand balance and the driving factors have rarely been considered or launched.

Based on healthy diet and national consumption standards, here, the LCC model and nutrient conversion model were used to reveal the change in the cereal and food calorie supply–demand balance in Uzbekistan during the period of 1995–2020. The study’s objectives are three-fold: (1) to characterize the production and dietary consumption structure of the inhabitants, (2) to analyze the cereal and food calorie supply and demand levels and variations under two different scenarios in Uzbekistan, and (3) to provide policy support for Uzbekistan to achieve national targets for food self-sufficiency. This paper explored food security, food production and consumption in Uzbekistan from the perspective of supply and demand, assisting decision making regarding agricultural development in a scientific manner. The results may help provide scientific support for food security and sustainable development in ecologically fragile regions globally.

## 2. Materials and Methods

### 2.1. Study Area

Uzbekistan is located in Central Asia and bordered by Kazakhstan to the west and north, Afghanistan and Turkmenistan to the south, and Kyrgyzstan and Tajikistan in the east (Figure 1). Uzbekistan has a total area of 4.49 × 10^5^ km^2^, of which about 4.02 × 10^4^ km^2^ (8.95%) is arable land, including temporary crops, temporary meadows and pastures, and temporary fallow land. The total population was 34.92 million in 2021, making it is the most populous country in Central Asia, and the population density is 77.7 person/km^2^. Uzbekistan has faced some issues, such as the rapidly shrinking Aral Sea, an immense cotton industry, huge deserts, advancing desertification, and concerns over potable water. Much of Uzbekistan’s cropland is irrigated. Due to the degradation of the limited arable land, it is vulnerable to climate change. Uzbekistan’s economy has rapidly developed, as the GDP continuously grew from 13.35 × 10^9^ USD in 1995 to 69.24 × 10^9^ USD in 2021, and the GDP in 2021 was 5.18-fold higher than that for 1995. Agriculture is the primary industry and provides a great contribution to the GDP. Wheat is the main cereal crop and Uzbekistan ranks among the global top consumers of bread and other wheat products.

The landform of the country is mainly plains, which accounts for 85% of the total land. It has a temperate continental climate, with long hot summers and a large difference in temperature between day and night. The annual precipitation in the plains is about 120–200 mm, while it is up to nearly 500–600 mm in the mountains. The arid climate makes the region suitable for growing melons and grapes. It only has inland rivers and the precipitation level is lower, leading to large desert areas. Its unique climatic characteristics make it a traditional agricultural country and an important vegetable-producing region in Central Asia. It exports large quantities of vegetables to neighboring countries such as Kazakhstan and Russia every year. Uzbekistan’s land use consists of cropland, grassland, forest, water, bare land, wetland, shrubland, impervious surface, and permanent snow or ice. Among them, the cropland area is 1.24 × 10^6^ km^2^, which accounts for 19.60% of the total land (Figure 1). The cropland area used for the cultivation of crops includes arable land and permanent crop land. This kind of land mainly provides agricultural plants such as cereals, fruits, and potatoes. In particular, cereals account for 46.33% of the total agricultural land. The grassland area is smaller than the cropland area, and it accounts for 5.92% of the total land. More importantly, livestock farming, which is based on grassland development, has a long history and is developing rapidly in Uzbekistan. Meat products make up a large proportion of the country’s production. Additionally, animal husbandry mainly consists of cattle, sheep, and poultry. Its primary animal products are beef, mutton, chicken, milk, and eggs. In 2019, the proportion of agriculture in the gross value added of industries was 26.9% (Figure 1). Agricultural land is used for cultivation of crops and animal husbandry.

### 2.2. Study Framework

As noted above, LCC is essentially a measure of the balance between food consumption and production, as well as evaluation of food supply–demand balance. This study focused on whether the limited land resources can meet the needs of the number of people living in the region. First, the effective calorie supply and dietary nutrition levels were estimated based on an analysis of the characteristics of land use and production with a food–calorie conversion model. Then, the different food demand and calorie requirement levels at different standards of living were assessed using an LCC model (Figure 2).

### 2.3. Research Methodology

#### 2.3.1. Food–Calorie Conversion Model

Each food has a different calorie value. The food supply and demand levels can be measured using the food–calorie conversion model:(1)$$Energy=\sum F_{i}\times Cal$$
where Energy is the calorie supply level, and the unit is kcal. F*_i_* represents the *i*th categories of food according to the quantity, and the unit is hectograms. Cal is the calorie contained in the *i*th categories of food, and the unit is kcal/100 g. The calorie contents of major food groups produced in Uzbekistan in the last 5 years are given in Table 1 for the purpose of estimating the effect of calorie intake on consumption.

#### 2.3.2. Cereal Supply–Demand Balance: LCC Model

According to the characteristics of food production in Uzbekistan, the following model was used to analyze the basic requirement of food demand and calorie:(2)$$LCC=\left\{\begin{array}{c} CLCC=C/C_{PC}\\ ELCC=E/E_{PC} \end{array}\right.$$
(3)$$LCCI=\left\{\begin{array}{c} CLCCI=Pa/CLCC\\ ELCCI=Pa/ELCC \end{array}\right.$$
where *CLCC* is the *LCC* estimating the population quantity cereal demand to be satisfied by cereal production. Cereals are plant-based foods and the primary constituents of most staple foods. According to the concept of land carrying capacity, *CLCC* is used to express the number of people that can be supported by cereal production. *ELCC* is the *LCC* estimating the quantity to be satisfied by calories supplemented by food production, C is the cereal production, *E* is the food calorie supply, *C_PC_* is the per capita cereal demand, and *E_PC_* is the per capita calorie demand. The quantities of cereals and calories required for maintaining the basic physiological activities of world residents are estimated to be 232 g/person/d and 2370 kcal/person/d, respectively [37]. Estimation of physical quantities is based on their waste, processing, seed, and feed. *Pa* is the current population size, *LCCI* is the *LCC* index, and *CLCCI* is an estimated *LCCI* based on cereal demand, which measures the degree of balance between the supply and demand of the population and grain. The relationship between supply and demand in Uzbekistan and its regions was analyzed using *CLCCI*. *ELCCI* is an estimated *LCCI* based on calorie demand, which measures the degree of balance between supply and demand of calories. *LCCI* values were classified into three levels and five sub-levels for describing the load on land resource (*LoL*) level (Table 2). Finally, *LoL* levels were divided by the *LCCI* value range.

#### 2.3.3. Definitions of Food Demand and Calorie Consumption Levels

*Cpcu* was analyzed according to the Uzbekistan cereal supplement under the Uzbekistan standard. The national consumption level of Uzbekistan was calculated based on the average supply quantity of cereal in 2015–2019. *Epcu* was analyzed according to the Uzbekistan calorie supplement under the Uzbekistan standard in 2015–2019 from FAO (Table 3). The number of people that can be supported by food production in Uzbekistan was calculated based on the per capita dietary energy consumption in Uzbekistan with the average of 3228.2 kcal, approximating 3200 kcal for the 5 years from 2015 to 2019. *Cpch* is the cereal intake in a healthy diet and *Epch* is the calorie intake in a healthy diet [37]. The calculation of total cereal and calorie consumption was based on the proportion of feed, seed, processing, waste and other uses of food in Uzbekistan. The quantities of cereal and calorie intake for maintaining the health of world residents were estimated to be 135 g/person/d and 4500 kcal/person/d, respectively. Based on Uzbekistan’s per capita dietary energy consumption of 3200 kcal/person/day for 1995–2019 and the healthy diet calorie provision standard of 4500 kcal/person/day, the number of people that can be supplied with energy from food production under both standards was calculated. The ratio of the actual to feedable population was used to analyze the relationship between supply and demand and its changes in Uzbekistan.

### 2.4. Data Sources

In this research, food refers to animal-based food and plant-based food. The animal-based food includes six sub-classes, namely buffaloes, sheep and goats, poultry birds, milk, eggs, and honey. The plant-based food comprises cereals, pulses, oil crops, potatoes, vegetables and fruits (Table 1). Among them, fruits include melons, pome fruits, stone fruits, tree nuts, subtropical fruits, berries, citrus fruits, and grapes. Vegetables consist of tomato, cucumbers, eggplant, pepper, carrot, bulb onions, garlic, cabbage, and mushrooms. Food production and consumption data for Uzbekistan from 1995 to 2020 were obtained from food balance sheets at FAO [38]. The size of the population, gross domestic production (GDP), and other economic data were obtained from the World Bank [39]. Land use data were extracted from the European Space Agency [35].

## 3. Results

### 3.1. Quantity Changes in Food Production

During 1995–2020, the production of plant-based food showed an increasing tendency as a whole across Uzbekistan, especially vegetables, fruits, starchy roots, and wheat (Figure 3a). Among them, the largest group of plant-based food was vegetables in 2020, with a production quantity of 99.04 × 10^5^ t, followed by fruits at 58.25 × 10^5^ t, and wheat at 61.58 × 10^5^ t. Statistical results showed that the fastest growing plant-based food was starchy roots in the last 25-years, whose production increased by more than 7 times, followed by vegetable, with a 3.63-fold increase, and wheat and maize, with a 3-fold increase. The increase in production for maize was smaller than that for wheat. In contrast, rice and barley production decreased (Figure 3a).

Next, the production of animal-based food in general showed an increasing trend in Uzbekistan during the period of 1995–2020 (Figure 3b). The largest group of animal-based food was milk in the last 25 years, and its production quantity reached 10.93 × 10^6^ t. This was followed by meat, at 58.25 × 10^5^ t, and eggs, at 43.61 × 10^4^ t. Since 1995, the fastest growing animal-based food was eggs, whose production increased by more than 6 times, followed by milk, with a 3-fold increase, and meat, with a 2.37-fold increase (Figure 3).

### 3.2. Structural Characteristics of Food Consumption

The diet of Uzbekistan is mainly based on grains, supplemented by milk, with vegetables and meat also playing an important part. According to the food consumption of Uzbekistan in 2019 (Figure 4), the per capita consumption of vegetables, milk, cereals, fruit, meat, and sugar was higher than that required for a healthy diet, while the per capita consumption of vegetable oils, fish, tree nuts and pulses was lower than that required for a healthy diet. In particular, the per capita consumption of vegetables was approximately 263.09 kg, which was much higher than the healthy diet standard of 126.47 kg. This was followed by milk, whose consumption (244.53 kg) was much higher than the healthy diet standard in 2019, with a difference of almost 126.14 kg. Due to the hot and dry climate and the long growing season, Uzbekistan has good-quality wheat and flour. Wheat is the main food crop in Uzbekistan, and the staple food of the Uzbek population is mainly made from wheat. The annual per capita consumption of cereals is about 191.36 kg, almost 60 kg more than the healthy diet standard. The consumption of pulses was 161.72 kg lower than the healthy diet standard, followed by tree nuts, at 8.02 kg (Figure 4). It is worth noting that the main meat ingredients are chicken, sheep and cattle, and most people do not eat pork due to their religious beliefs.

In terms of nutrient sources, Uzbekistan’s food consumption calories were mainly derived from cereals, with dairy foods as a supplement. Cereals, dairy foods and vegetable oils contributed 46.55%, 12.83% and 7.41% of calories, respectively; meat, sweeteners and vegetables accounted for more than 5% of total calorie consumption; while tree nuts, eggs, fish and pulses accounted for less than 1% (Figure 5a). Comparing the proportion of national calories from food supply sources between 1995 and 2019, the ratio of cereals and vegetable oils decreased, while the ratio of dairy foods, starchy roots, vegetables, fruits and sweeteners increased. The proportion of calories from cereals decreased by 13.14%, followed by vegetable oils, which decreased by 5.27%. The proportion of calories from dairy foods increased by 4.06%, that from starchy roots, vegetables and fruits increased by almost 3%, and that from sweeteners increased by 2.08% (Figure 5a). Comparing the proportion of calories from food supply sources in Uzbekistan with healthy diets in 2019 revealed differences in the structure of consumption in Uzbekistan from that of healthy diets. The proportion of calories from cereals, dairy foods, meat, starchy roots, vegetables, and sweeteners was higher than the healthy diet standard in Uzbekistan, while the proportion of calories from pulses, vegetable oils, tree nuts and fish was lower than the healthy diet standard. The differences of proportion of calories from fruits, animal fats and eggs was less than 1%. Concretely speaking, the caloric portion of cereals in Uzbekistan was significantly higher than the healthy diet standard, with a difference of 14.15%, following by dairy foods, with a difference of 6%, and meat, with a difference of 3.70%. The percentage of calories from pulses in Uzbekistan was obviously below the healthy diet standard, with a difference of 16.86%, followed by vegetable oils, with a difference of 9.13%, and finally tree nuts, with a difference of 5.39% (Figure 5).

Moreover, calorie supply in Uzbekistan decreased first and then increased (Figure 6), from 2655 kcal/capita/day in 1995 to 1892 kcal/capita/day in 2002 and then back to 3219 kcal/capita/day. In 2019, plant-based foods accounted for about 80% of calories, and were the main source of food calories. In terms of plant-based food groups, except for cereals and vegetable oils, the calorie supply increased in all food groups. Additionally, the plant-based food calorie percentage declined slightly, from 82.75% to 76.64%. However, animal-based food consumption increased from 458 kcal/capita/day in 1995 to 752 kcal/capita/day in 2019. Due to the supply of milk, the animal-based food consumption increased from 17.25% to 23.36% of the total food consumption.

### 3.3. Evaluation of the Relationship between Supply and Demand Based on Cereal Demand

From 1995 to 2020, CLCC_UZ_ showed fluctuating growth in Uzbekistan. CLCC_UZ_ was 17.8 × 10^6^ people in 1995, well below the actual population of 22.79 × 10^6^, showing that the cereal production of the country was unable to meet the needs of the population until 2001. As cereal yield increased, the CLCC_UZ_ began to exceed the population in 2002. Cereal supply and demand were in balance and began to be in surplus. The difference between the CLCC_UZ_ and the actual population reached a maximum of 10.68 × 10^6^ in 2016. Due to declining cereal production and population growth, the CLCC_UZ_ decreased slightly from 2017 onwards, and the gap between CLCC_UZ_ and actual population narrowed, but the CLCC_UZ_ still exceeded the population size except in 2018. In 2020, the CLCC_UZ_ was 37.45 × 10^6^ people in Uzbekistan, which was more than the population of 35.23 × 10^6^ (Figure 7a). In the last 25 years, CLCC_HD_ was higher than population consistently under the healthy diet consumption standard, and the trend was consistent with the Uzbekistan standard. CLCC_HD_ grew from 25.06 × 10^6^ to 52.75 × 10^6^ people. Due to the lower standard of 135 kg/p/year, the CLCC_HD_ was higher than CLCC_UZ_ (Figure 7b).

During the same time period, CLCCI showed a fluctuating decline in the Uzbekistan and healthy diet standards. The CLCCI decreased from 1995 to 2020, and reached its lowest level in 2009 with a surplus of cereals. However, the CLCCI increased from 2017 to 2020 due to population growth and unstable cereal production, with slight tension in cereal supply–demand relationship. CLCCI_UZ_ was over 1.125 in 1995 and in a state of overload until 1997. From 1998 to 2001, except for 2000, CLCCI_UZ_ was between 0.875 and 1.125, indicating a balanced supply and demand relationship for human cereals. Subsequently, the CLCC began to exceed the population from 2002 to 2016, and the CLCCI_UZ_ was between 0.5 and 0.875, implying surplus in the supply and demand relationship for human cereals. Then, CLCC_UZ_ slightly exceeded the population size, with CLCCI_UZ_ between 0.875 and 1.125 since 2017, suggesting that the cereal supply and demand relationship was in balance (Figure 8a). In the first place, CLCC_HD_ was slightly higher than the population size in 1995 and 1996 and CLCCI_HD_ was between 0.875 and 1.125, with a balanced supply and demand relationship for cereals. Moreover, CLCCI_HD_ was higher than 0.5 and lower than 0.875 until last year, with a surplus of cereals (Figure 8b).

### 3.4. Evaluation of the Relationship between Supply and Demand Based on Dietary Energy Demand

In this study, the food supply capacity gradually increased from 1995 to 2020 in Uzbekistan, exceeding the actual population in 2002 in the caloric consumption standard. In 1995, ELCC_UZ_ was 23.92 million people, which was 1.13 million more than the actual population, and calorie supply of food could meet demand. From 2002 to 2015, ELCC_UZ_ increased significantly, and the gap between it and the population was the largest in 2015, at 12.23 million. Later, ELCC_UZ_ declined until 2018 and then went up again, but remained higher than the actual population. As of 2020, ELCC_UZ_ was 42.44 million people, exceeding the population by 8.21 million (Figure 9a). For healthy diet consumption standard, ELCC_HD_ grew from 17.01 million to 30.18 million persons and was lower than ELCC_UZ_ for the higher standard of 4500 kcal/p/y. From 1995 to 2005, ELCC_HD_ was lower than the population, and the largest difference in 2000 was 8.40 million people. With the calorie supply increase, ELCC_HD_ was close to the population from 2006 to 2016, with minimum gap of 0.51 million people in 2015. Due to the reduction in calorie supply and the growth of the population since 2016, the population exceeded the ELCC_HD_ and food calorie supply was not adequate to feed the population (Figure 9b).

Since 1995, ELCCI developed in a positive and volatile trend (Figure 10). The results showed that ELCCI_UZ_ ranged from 0.875 in 1995 to 1.125 in 2002, indicating that the relationship between food supply and demand was in balance. ELCCI_UZ_ was between 0.5 and 0.875 from 2002 to 2020, except for 2018, indicating surplus in the food supply–demand relationship (Figure 10a). For the healthy diet consumption standard, ELCCI_HD_ exceeded 1.125 during 1995-2005 and then even exceeded 1.5 from 1996 to 2000, implying that with the food supply–demand relationship changed from overloaded to severely overloaded. ELCCI_HD_ was less than 1.125 and higher than 0.875 until 2016 with the increase in calorie supply, and the relationship between food supply and demand was balanced. Finally, due to the reduction in calorie supply and the growth of the population, ELCCI_HD_ was between 1.125 and 1.5 in the last 4 years, indicating that food supply and demand was overloaded (Figure 10b).

## 4. Discussion

### 4.1. Healthy Diet Standard

Typically, the healthy diet standard proposes recommended food intakes and calorie intakes. When food is produced, it is used not only for consumption but also for feed, seeds, processing, waste, and other uses (e.g., tourism consumption, etc.) It is necessary to convert the intake into consumption under the healthy diet standard. The various utilizations of food calories and their consumption under the healthy diet standard were calculated as percentages of each food type and its utilizations in Uzbekistan in the present year (average level of 2015–2019 in FAO). Specific proportions of food groups in different utilizations and intake are shown in Table 4. Cereal consumption and calorie consumption standards for Uzbekistan are based on the average values of food supply quantity (kg/capita/yr) of cereals and the total food supply (kcal/capita/day) for the last five years (2015–2019) in FAO food balance. Furthermore, the daily total food supply is converted into annual quantity.

There is a gap between the FAO food balance sheet and the healthy diet standard. This is because the FAO food balance sheet counts the total amounts of primary crops and their secondary products (e.g., wheat and flour), while the healthy diet standard considers only the number of primary crops. Future research will need to further explore matching indicators and following the principles of scientificity and comparability.

### 4.2. Impact of Structure of Food Import and Export on Food Supply–Demand Balance

From the perspective of Uzbekistan’s own supply, its cereal output could not meet the consumption demand in the last 25 years. Considering the import and export of food, we further discussed the balance of food supply and demand in Uzbekistan under the open system, combined with factors including the food conversion factor, intake, and loss.

The main imported foodstuff is wheat products in Uzbekistan, which declined from 1995 to 2004 and then rose rapidly, from 1506 × 10^3^ t to 2787 × 10^3^ t, an increase of nearly 2-fold. Flour-based wheat products are imported due to the poor quality of flour produced in the country. The importation of sugar rose steadily, from 169 × 10^3^ t to 674 × 10^3^ t, a nearly 4-fold increase. Exports of potatoes and vegetable oils showed an overall growth trend (Figure 11). From 1995 to 2019, Uzbekistan’s crop exports were mainly fruits, wheat, and vegetables, showing an upward trend, except for cotton. Vegetable exports increased by three times from 18.68 × 10^3^ t in 1995 to 649 × 10^3^ t in 2019. Fruit exports increased by nearly 11 times from 94 × 10^3^ t to 1038 × 10^3^ t, of which grape exports accounted for nearly half in recent years. Vegetable exports increased by three times from 209 × 10^3^ t in 1995 to 649 × 10^3^ t in 2019. Exports of pulses went from zero in 1995 to 769 × 10^3^ t in 2019. Uzbekistan’s cotton lint exports declined from 1025 × 10^3^ t to 159 × 10^3^ t (Figure 11).

As can be seen from Uzbekistan’s imports and exports, due to the scarcity of water resources and environmental degradation, Uzbekistan’s exports have shifted from cotton to fruit and vegetables. Imports have been dominated by wheat, supplemented by sugar. The lack of production of high-calorie cereal foods and the extensive cultivation of export-oriented vegetables and fruits with high added value but low calorie content has led to a healthy diet scenario where calorie supply cannot meet domestic demand.

Furthermore, the supply capacity and supply–demand relationship of food in Uzbekistan were explored under an open system considering food production and imports. Compared with the production-only ELCC, supply capacity rose by an average of 531.19 thousand people per year under the open system. The relationship between supply and demand has improved significantly in the last decade, being in balance with the healthy diet standard and remaining in surplus for the last 15 years under the national diet standard (Figure 12). However, the production data are derived from FAO production, which considers the primary agricultural product, and the import data are derived from the FAO food balance sheet, which considers both primary and secondary agricultural products. Notably, only the main crops are considered for the convenience of calculation. Therefore, the calorie supply capacity is lower than the value obtained by only considering production crop subcategories in the period 1995–2007. Moreover, the open system status of food subcategories can make the research more scientific and applicable.

## 5. Conclusions

Based on the land resource carrying capacity (LCC) model, here, data from the Food and Agriculture Organization (FAO) and the World Bank were combined to explore the levels of cereals and calories in the food supply and the food supply–demand relationship in Uzbekistan and its changing trends during the period of 1995–2020. Due to the hot and dry climate, the large temperature difference between day and night and the long growing time, Uzbekistan has a high yield of vegetables and fruits. Crop production is dominated by vegetables and fruits in Uzbekistan, with wheat as the main food crop. Meanwhile, production of meat, eggs and milk has shown an overall increase. Meat is dominated by bovines, accounting for 80% of meat production, followed by lamb, and a relatively small proportion of other meat. The structure of production determines the structure of food consumption in Uzbekistan. Uzbekistan’s calorie intake is mainly from cereals, with milk as a supplement and insufficient intake of pulses, vegetable oils, tree nuts, and fish to meet healthy dietary standards.

In general, it was found that the fluctuating growth of CLCC in Uzbekistan from 1995 to 2020, both in the Uzbekistan scenario and in the healthy diet scenario, has resulted in a balanced and surplus supply and demand for human cereals over the last decade. The domestic cereal production meets the needs of the domestic population. Under the Uzbekistan consumption scenario, the CLCCI shifted from surplus to balance in 2017–2020, with a tightening of the cereal supply–demand relationship. From 1995 to 2020, the ELCC in Uzbekistan continued to increase. Under the Uzbekistan calorie consumption scenario, the carrying state shifted from equilibrium to surplus to meet the demand for food calories. However, under the healthy diet scenario, the carrying state shifted from overload to balance and then to overload in 2017–2020.

In addition, the production mix has shifted from wheat to mainly vegetables and fruits in Uzbekistan, which are used as exports to generate higher added value. Consumption is dominated by high-calorie cereals, which have decreased in the last 25 years but are still above healthy dietary levels, followed by milk. Cereal production meets the needs of the population in both scenarios, with a slight calorie deficit at healthy dietary standards. The structure of its production has improved significantly, both to meet cereal demand and to generate value. The consumption structure differs considerably according to healthy dietary standards and is dominated by its livestock and wheat production. The pulses produced are largely used for feed and their consumption is minimal, which may cause nutrient deficiencies and lead to disease, so imports of pulses should be increased moderately. These findings may help the government of Uzbekistan to adjust the quantity of crops produced and the structure of imports and exports to achieve a balance between quality and quantity, and to shift their consumption structure towards a healthier diet. Further research should be undertaken to investigate the impact of imported primary processed crops on the nutritional supply and demand balance for food and the magnitude of their contribution to reducing pressure on land.

## Figures and Tables

**Figure 1 foods-12-02065-f001:**
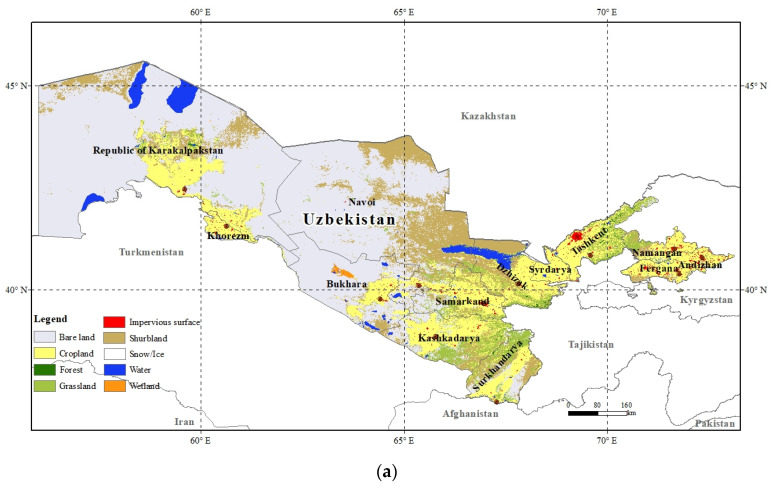
Location of Uzbekistan showing the (**a**) land use [35] and (**b**) population [36] in 2020.

**Figure 2 foods-12-02065-f002:**
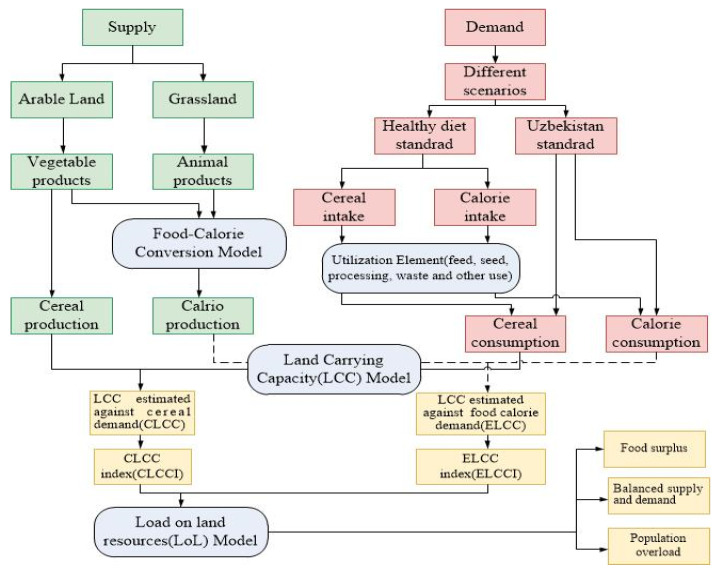
Study framework and approach.

**Figure 3 foods-12-02065-f003:**
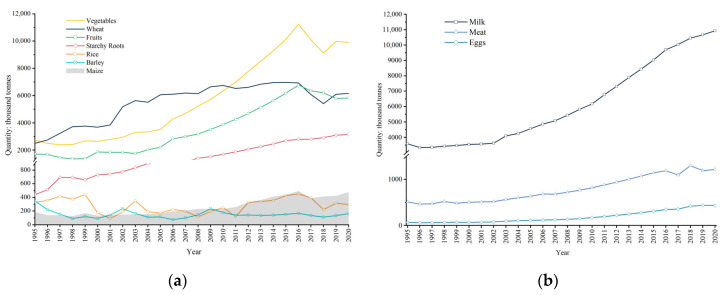
Annual production of (**a**) plant-based food and (**b**) animal-based food in Uzbekistan from 1995 to 2020.

**Figure 4 foods-12-02065-f004:**
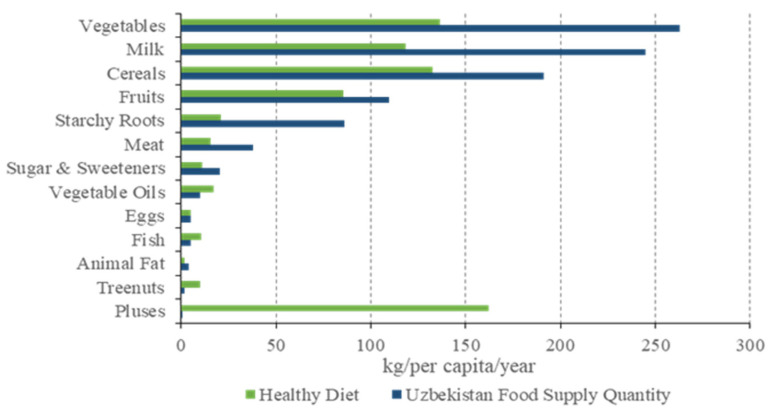
Food consumption of Uzbekistan in 2019 in different scenarios of the healthy diet standard and national standard.

**Figure 5 foods-12-02065-f005:**
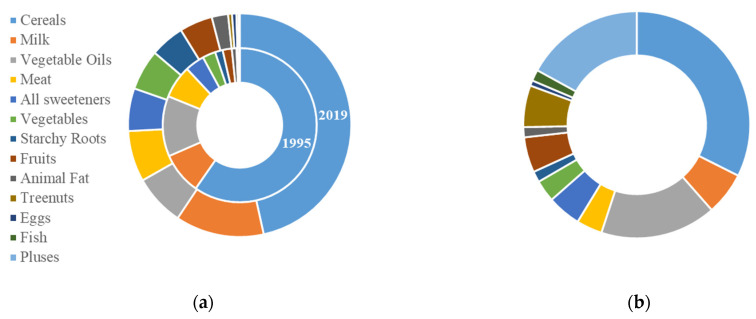
Structural characteristics of (**a**) food consumption in 1995 and 2019 as well as (**b**) a healthy diet in terms of calorie share of major foods (%) in Uzbekistan.

**Figure 6 foods-12-02065-f006:**
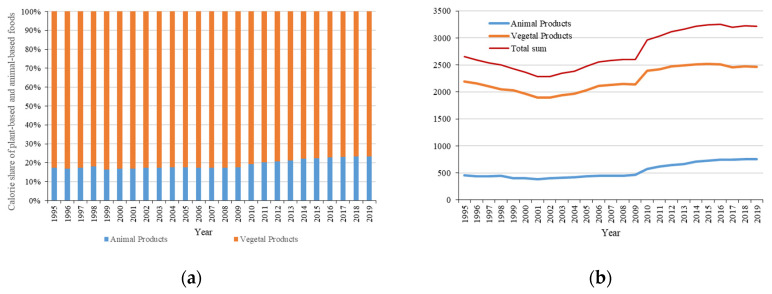
The calories of (**a**) vegetable and animal products and (**b**) the variety of food consumption in Uzbekistan.

**Figure 7 foods-12-02065-f007:**
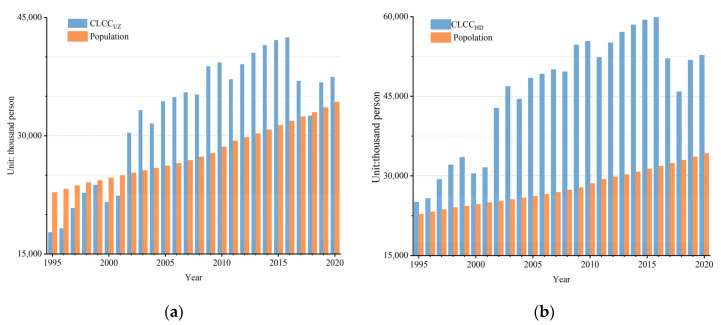
Annual changes in CLCC under the (**a**) national and (**b**) healthy diet standards in Uzbekistan during the period of 1995–2020.

**Figure 8 foods-12-02065-f008:**
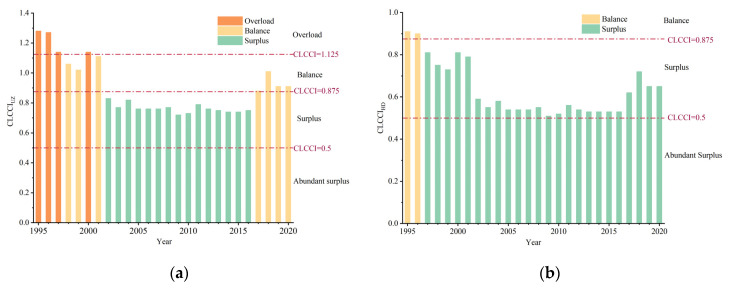
Annual changes in CLCCI under the (**a**) national and (**b**) healthy diet standards in Uzbekistan during the period of 1995–2020.

**Figure 9 foods-12-02065-f009:**
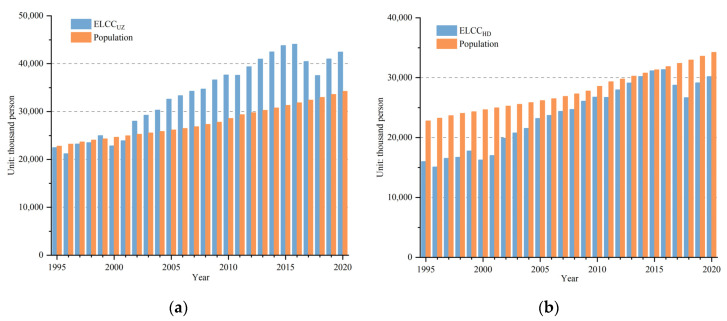
ELCC under (**a**) Uzbekistan and (**b**) healthy diet standards in Uzbekistan.

**Figure 10 foods-12-02065-f010:**
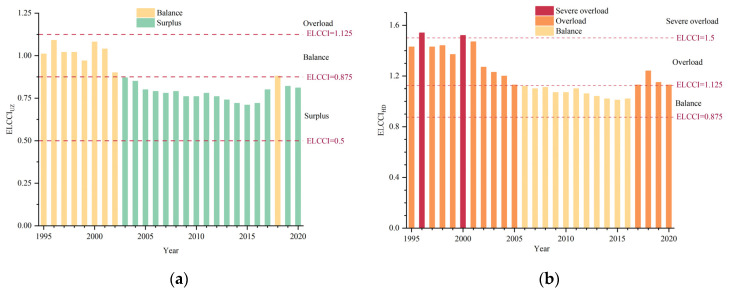
ELCCI under (**a**) Uzbekistan and (**b**) healthy diet standards in Uzbekistan.

**Figure 11 foods-12-02065-f011:**
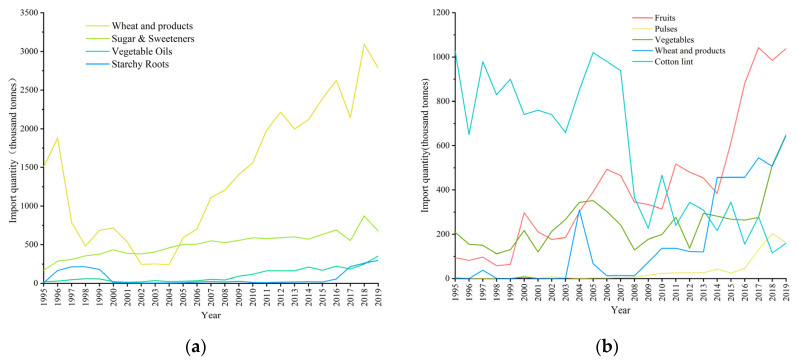
Food (**a**) import quantity and (**b**) export quantity in Uzbekistan.

**Figure 12 foods-12-02065-f012:**
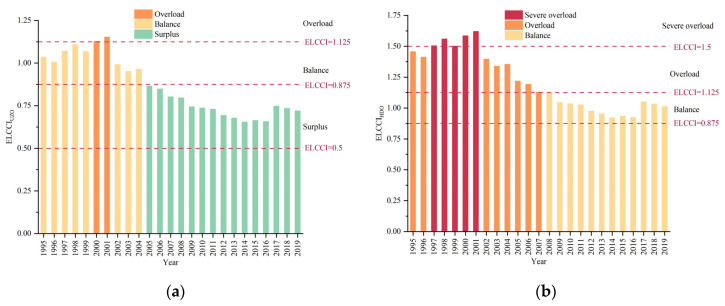
ELCCI under (**a**) Uzbekistan and (**b**) healthy diet standards in Uzbekistan open system.

**Table 1 foods-12-02065-t001:** Food–calorie conversion parameters for major categories of foods.

Plant-Based Foods	Calories (kcal/100 g)	Animal-Based Food	Calories (kcal/100 g)
Wheat	334	Meat	122
Rice	280	Eggs	139
Maize	356	Milk	61
Barley	332	Sugar and sweeteners (honey)	298
Starchy roots	67		
Sugar crops	70		
Pulses	343		
Fruit	39		
Vegetables	25		
Tree nuts	275		
Oil crops	884		

**Table 2 foods-12-02065-t002:** Classification of LoL levels and sub-levels according to the value of LCCI.

LoL Level	LoL Sub-Level	LCCI Value Range
Food surplus	Abundant surplus	<0.5
	Surplus	0.5–0.875
Balanced supply and demand	Balance	0.875–1.125
Population overload	Overload	1.125–1.5
	Severe overload	>1.5

**Table 3 foods-12-02065-t003:** Cereal and calorie supplement levels in Uzbekistan at different standards of living.

Standard of Living	Cereals (kg/Person/y)	Calories (kcal/Person/d)
Healthy Diet Standard	135.00	4500.00
Uzbekistan standard	190.00	3200.00

**Table 4 foods-12-02065-t004:** Food groups in different utilizations and intake under the healthy diet standard.

	Macronutrient Intake (g/day)	Caloric Intake (kcal/day)	Feed (%)	Seed (%)	Processing (%)	Waste (%)	Other use (%)
Cereals	232	811	42.15%	4.46%	4.58%	5.40%	0.26%
Starchy Roots	50	39	0.60%	11.61%	0.00%	1.15%	0.00%
Vegetables	300	78	17.63%	0.00%	0.04%	7.09%	0.00%
Fruits	200	126	4.41%	0.00%	6.19%	6.58%	0.00%
Milk	250	153	3.36%	0.00%	26.12%	0.26%	0.00%
Meat	43	92	0.00%	0.00%	0.00%	0.00%	0.00%
Eggs	13	19	0.00%	1.48%	0.00%	8.46%	64.27%
Fish	28	40	4.06%	0.00%	0.00%	0.00%	0.00%
Pluses	100	426	316.87%	6.02%	0.00%	21.69%	0.00%
Treenuts	25	149	0.00%	0.00%	0.00%	6.94%	0.00%
Vegetable Oils	46.8	414	0.00%	0.00%	0.00%	0.00%	4.75%
Animal Fat	5	26	0.00%	0.00%	0.00%	7.81%	61.46%
Added sugars	31	120	0.00%	0.00%	0.00%	0.15%	0.00%

## Data Availability

The data presented in this study are available request to corresponding authors.

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
