# Peer review of "Spatiotemporal Characteristics of Food Supply–Demand Balance in Uzbekistan under Different Scenarios"

_foods, 2023, doi:10.3390/foods12102065_

Round 1

Reviewer 1 Report

The topic under study is intriguing, and the paper's objective is evident. The article adheres to journal guidelines and is well-organized. The research methodology is accurate and is explained thoroughly in both the text and diagrams. In general, the paper is a valuable contribution to the scientific community, particularly for Uzbekistan's planners seeking to establish long-term food supply-demand equilibrium in various scenarios throughout the country. Other developing nations may adopt the research methodology applicable to comparable subjects. The final two sentences of the conclusions offer a broader perspective on how to use the research findings presented in the paper and direct future research efforts.

Below is suggestion how to improve manuscript:

1.      Abstract can be shortened and please without acronyms. I suggest to move details from abstract into introduction and conclusions and rewrite (abstract) as follows:’The food supply-demand balance is an everlasting topic for many countries, especially for sustainable development, particularly in developing countries. Using the land resource carrying capacity model, we analyzed cereal and calorie supply and demand in Uzbekistan from 1995 to 2020. Despite increased demand for cereal and calories, unstable crop production has led to volatile growth patterns. The carrying capacity of crop land resources under national consumption standard shifted from overload to surplus to balance, while the healthy diet standard moved from balance to surplus. The calorific equivalent land resource carrying capacity under consumption standard fluctuated but shifted from balance to surplus, while the healthy diet standard remained deficient. Our findings can help guide a sustainable production and consumption strategy in Uzbekistan.’

2.      Check typos and do some rewriting to clarify relevance of results. I also suggest to rewrite text in lines 103-107 to be grammar correct.

Author Response

Dear Reviewer,

Thank you very much for your time involved in reviewing the manuscript and your very encouraging comments on the merits.

On behalf of all the contributing authors, I would like to express our sincere appreciations of your letter and reviewers’ constructive comments concerning our article entitled “Spatio-temporal characteristics of food supply-demand balance in Uzbekistan under different scenarios” (Manuscript No: 2371180). These comments are all valuable and helpful for improving our article. According to reviewer’s comments, we have made extensive modifications to our manuscript and supplemented extra data to make our results convincing. In this revised version, changes to our manuscript were all marked within the document. Point-by-point responses to the nice associate editor and two nice reviewers are listed below this letter.

We feel great thanks for your professional review work on our article. As you are concerned, there are several problems that need to be addressed. According to your nice suggestions, we have made extensive corrections to our previous draft, the detailed corrections are listed below.

Point 1: Abstract can be shortened and please without acronyms. I suggest to move details from abstract into introduction and conclusions and rewrite (abstract) as follows:’The food supply-demand balance is an everlasting topic for many countries, especially for sustainable development, particularly in developing countries. Using the land resource carrying capacity model, we analyzed cereal and calorie supply and demand in Uzbekistan from 1995 to 2020. Despite increased demand for cereal and calories, unstable crop production has led to volatile growth patterns. The carrying capacity of crop land resources under national consumption standard shifted from overload to surplus to balance, while the healthy diet standard moved from balance to surplus. The calorific equivalent land resource carrying capacity under consumption standard fluctuated but shifted from balance to surplus, while the healthy diet standard remained deficient. Our findings can help guide a sustainable production and consumption strategy in Uzbekistan.’

Response 1: First, we think it's a good proposal that the abstract be rewritten. So we have rewritten the abstract and changed the acronym to the full name, the specific changes are in lines 11-35 of the article. Also listed in the accompanying table are abbreviations and their full names through Appendix to make it easier for readers to understand. The changes have been marked with a revision.

Point 2: Check typos and do some rewriting to clarify relevance of results. I also suggest to rewrite text in lines 103-107 to be grammar correct.

Response 2: Due to a grammatical error, make grammatical corrections to original lines 103-107, in lines 107-118 of the article. We tried our best to improve the manuscript and made some changes to the manuscript. These changes will not influence the content and framework of the manuscript. And here we did not list the changes but marked up using the “Track Changes” in the revised manuscript.Errors in terminology and grammar have been corrected.

We tried our best to improve the manuscript and made changes marked in revised manuscript which will not influence the content and framework of the manuscript. We appreciate for reviewer’ s warm work earnestly, and hope the correction will meet with approval. Once again, thank you very much for your comments and suggestions.

Reviewer 2 Report

In the Abstract, apart from providing a lot of information forming the background of the considerations and elements of the research method, it was also worth clearly formulating the purpose of the research / research study. The authors wrote a lot about what they did, but it would be worth considering the purpose of the research undertaken on these issues.

At the end of the last paragraph in the Introduction, the authors wrote what they focused on when developing the topic, but such wording is difficult to consider as a clearly stated research goal / study. I suggest that the purpose of the research / study should be written in a more clear and unambiguous way. The purpose of the research/study was … In addition, I suggest writing what was the cognitive (scientific) and what was the utilitarian (useful) purpose of the study. Thanks to this, in the Conclusions it will be easier to refer to these goals and determine whether they have been achieved and what are the prospects for further research to be carried out in a given thematic scope.

Before stating the objectives of the research study, it would be worth formulating the research problem. I think that based on the review of the state of knowledge presented in the Introduction of the article, one can easily formulate a research problem. I suggest that in the summary of the state of the art simply write the sentence: "The research problem is ...". The research problem can be linked to the presentation of a gap in the current state of knowledge, which is then translated into the formulation of the research goal / study.

In subchapter 2.1 (Study Area) it would be worth specifying the year for which the data on the population of Uzbekistan residents and population density refer. Taking into account the population of inhabitants in the analysis / model, taking into account the formula (3), it would be worth paying attention to the fact that the population may be a variable value. How does the model take into account emigration / immigration, which affects the state of the population in the country?

The approach to the study presented in Figure 2 shows that the supplies of plant raw materials, including those for animal production, come only from domestic resources, because Arable Land and Grassland are given. Does the supply balance not include imports of grain and other raw materials of plant origin? Such imports balance the resources of plant raw materials in the national food balance. It is worth mentioning in the Materials and Methods chapter.

Have the risk elements been taken into account in the research / study, including, for example, a possible reduction in crop yields caused by adverse weather phenomena and other factors? Thanks to the risk assessment, the proposed model will be more credible.

I am missing units in formula (1) and its description. Although calories appear in the description, the Fi parameter already informs about the amount of a given food category; but it is not known in which units the amount of a given food category is taken into account. It is true that formula (1) presents only a model of approach to research, but in my opinion it would be worth mentioning units.

Considering the energy associated with food in the analysis, it would be worth developing the question of what energy is taken into account. Is it rated energy (at billing level), usable energy, or cumulative energy? This is especially important in the case of animal products, for which the cumulated energy is higher than it was included in the model.

In some parts of the analysis, the end year 2019 is taken into account, and in some parts 2020. It would be worth explaining where these differences come from (probably due to the availability of data) and how they affect the results of the analysis / study.

In Figure 5, it would be worth specifying the units in which the elements of the structure of consumption and a healthy diet were given. Of course, you can guess the units, but it's better to write this information.

I think on the ordinate (y) axis in Figure 6b you don't need to write: "Unit:". On the ordinate axis (y) in Figure 6a, it is worth writing the title of this axis. You can guess that it's about the percentage share of two types of products, but it needs to be spelled out.

It would be worth reviewing the content of the article carefully and correcting minor linguistic errors. For example, the legend in Figure 6b reads "Grand Total". In my opinion it should be "Total sum". It is worth reviewing the article in terms of the correctness of the language terms used.

In the captions of figures 7-10, instead of acronyms, it would be worthwhile to provide the full names of the considered indicators. In the current version, it is very difficult to read and interpret individual graphs and their descriptions if only acronyms are given. The more so that the graphs (on the ordinate axis - y) also show only acronyms, so full understanding of the content of the figure requires searching for the full name of a given acronym in the text, which is a significant difficulty for the reader.

I would like to ask about the error in the model analysis adopted by the authors. Did the authors take this error into account? I request information.

Do the presented model studies allow to determine the prospects for balancing food supply and demand in Uzbekistan? How long a perspective can be developed based on the conducted research study? It is worth writing about it in the article.

Did the authors take into account in the analysis the nutritional models related to the changing preferences in terms of the quantity and quality of the components of the diet of the inhabitants of Uzbekistan? It's worth writing about.

I think that in some part of the article, at the beginning or at the end, it is worth making a list of all the acronyms used in the study. There are so many acronyms that it is very difficult to read and analyze the content of the article, so it is worth listing all the acronyms in one place. 

The article requires - in my opinion - minor language corrections. A large number of acronyms makes it difficult to read the text, so I suggest listing all acronyms together with their full name in one place. 

Author Response

Dear Reviewer,

Thank you very much for your time involved in reviewing the manuscript and your very encouraging comments on the merits.

On behalf of all the contributing authors, I would like to express our sincere appreciations of your letter and reviewers’ constructive comments concerning our article entitled “Spatio-temporal characteristics of food supply-demand balance in Uzbekistan under different scenarios” (Manuscript No: 2371180). These comments are all valuable and helpful for improving our article. According to reviewer’s comments, we have made extensive modifications to our manuscript and supplemented extra data to make our results convincing. In this revised version, changes to our manuscript were all marked within the document. Point-by-point responses to the nice associate editor and two nice reviewers are listed below this letter.

Point 1: In the Abstract, apart from providing a lot of information forming the background of the considerations and elements of the research method, it was also worth clearly formulating the purpose of the research / research study. The authors wrote a lot about what they did, but it would be worth considering the purpose of the research undertaken on these issues.

Response 1: We think the suggestion to add the purpose of the study to the abstract is a good one, and have rewritten the abstract and added the purpose of the study to appear in lines 33-35.

Point 2: At the end of the last paragraph in the Introduction, the authors wrote what they focused on when developing the topic, but such wording is difficult to consider as a clearly stated research goal / study. I suggest that the purpose of the research / study should be written in a more clear and unambiguous way. The purpose of the research/study was … In addition, I suggest writing what was the cognitive (scientific) and what was the utilitarian (useful) purpose of the study. Thanks to this, in the Conclusions it will be easier to refer to these goals and determine whether they have been achieved and what are the prospects for further research to be carried out in a given thematic scope.

Response 2: At the end of the last paragraph of the 'Introduction', which was of concern when developing the topic, the wording was not considered clear enough. We have made the purpose of the research/study clearer and more explicit. Change the purpose of the study to " The study has three-fold objectives: (1) to characterize the production and dietary consumption structure of the inhabitants, (2) to analyze cereal and food calorie supply and demand levels and variations under two different scenarios in Uzbekistan, and (3) to provide policy support for Uzbekistan to achieve national targets for food self-sufficiency ".

Point 3: Before stating the objectives of the research study, it would be worth formulating the research problem. I think that based on the review of the state of knowledge presented in the Introduction of the article, one can easily formulate a research problem. I suggest that in the summary of the state of the art simply write the sentence: "The research problem is ...". The research problem can be linked to the presentation of a gap in the current state of knowledge, which is then translated into the formulation of the research goal / study .

Response 3: In the summary of the state of the art, the research question is emphasised and translated into a statement of the research objectives. In the summary of the state of the art, the research question is emphasised and translated into a statement of the research objectives. In lines 82-87 of the manuscript " However, the research problem is that the study of food production and consumption still lacks a suitable and unique computational standard for assessing different types of food products in order to gain a clearer understanding of local food production and consumption, especially in developing countries like Uzbekistan. LCC model can better link food production and consumption to explore the supply and demand balance of national food consumption and ensure national food security."

Point 4: In subchapter 2.1 (Study Area) it would be worth specifying the year for which the data on the population of Uzbekistan residents and population density refer. Taking into account the population of inhabitants in the analysis / model, taking into account the formula (3), it would be worth paying attention to the fact that the population may be a variable value. How does the model take into account emigration / immigration, which affects the state of the population in the country?

Response 4: Population and density are applied to the 2021 data, increasing at line 125. Population is indeed a variable in the model, but this paper considers the total population of Uzbekistan, which has a net migration of -39 thousand people in 2021, only 0.1% of the population, relatively small number of migrants whose impact on the results of this manuscript is small, and this manuscript does not consider national migration or inter-regional migration.

Point 5: The approach to the study presented in Figure 2 shows that the supplies of plant raw materials, including those for animal production, come only from domestic resources, because Arable Land and Grassland are given. Does the supply balance not include imports of grain and other raw materials of plant origin? Such imports balance the resources of plant raw materials in the national food balance. It is worth mentioning in the Materials and Methods chapter .

Response 5: For whether the supply-demand balance excludes imports of cereals and other plant raw materials, the analysis in results does not take into account imports and exports, but the discussion that can be met with national production, analysising the supply and demand balance for cereals and food is carried out in discussion in conjunction with national imports and exports.

Point 6: Have the risk elements been taken into account in the research / study, including, for example, a possible reduction in crop yields caused by adverse weather phenomena and other factors? Thanks to the risk assessment, the proposed model will be more credible.

Response 6: Among the risk factors considered in the model, the risk issues facing Uzbekistan (such as rapidly shrinking Aral Sea, an immense cotton industry, huge deserts, advancing desertification, and concerns over potable water) are mentioned in the study area. The main objective of this manuscript is to focus on changes in supply and demand and to provide policy support for Uzbekistan to achieve its national goal of food self-sufficiency.

Point 7: I am missing units in formula (1) and its description. Although calories appear in the description, the Fi parameter already informs about the amount of a given food category; but it is not known in which units the amount of a given food category is taken into account. It is true that formula (1) presents only a model of approach to research, but in my opinion it would be worth mentioning units.

Response 7: Add the units for each indicator in equation (1), lines 174-179 on page 6 of the manuscript.

Point 8: Considering the energy associated with food in the analysis, it would be worth developing the question of what energy is taken into account. Is it rated energy (at billing level), usable energy, or cumulative energy? This is especially important in the case of animal products, for which the cumulated energy is higher than it was included in the model.

Response 8: The energy associated with primary food consumed by humans is considered in the analysis as rated energy, and the accumulated energy from animal food consumption does not have to be considered.

Point 9: In some parts of the analysis, the end year 2019 is taken into account, and in some parts 2020. It would be worth explaining where these differences come from (probably due to the availability of data) and how they affect the results of the analysis / study.

Response 9: For a portion of the data use 2019 as the end year and for a portion use 2020 as the end year. In this manuscript, food production data can reach 2020 and food consumption data to 2019. For consumption the main analysis is the current state of consumption and the analysis of consumption demand through the current state of consumption. As food consumption has not changed much in recent years, the analysis is done for the last 5 years from 2015-2019 to represent the current state of consumption, which has very little impact on the results.

Point 10: In Figure 5, it would be worth specifying the units in which the elements of the structure of consumption and a healthy diet were given. Of course, you can guess the units, but it's better to write this information.

Response 10: In Figure 5, due to the lack of units for the elements of consumption structure and healthy eating, we made changes in the title of Figure 5 and added unit information in rows 296-297.

Point 11: I think on the ordinate (y) axis in Figure 6b you don't need to write: "Unit:". On the ordinate axis (y) in Figure 6a, it is worth writing the title of this axis. You can guess that it's about the percentage share of two types of products, but it needs to be spelled out .

It would be worth reviewing the content of the article carefully and correcting minor linguistic errors. For example, the legend in Figure 6b reads "Grand Total". In my opinion it should be "Total sum". It is worth reviewing the article in terms of the correctness of the language terms used .

Response 11: Based on your opinion that on the ordinal (y) axis in Figure 6b it is not necessary to write: "Units:" and on the ordinal axis (y) in Figure 6a, labeled with the axis title, the legend in Figure 6b is the opinion of "Total sum". Corrected Figure 6 as you requested, on page 10 of the article.

Point 12: In the captions of figures 7-10, instead of acronyms, it would be worthwhile to provide the full names of the considered indicators. In the current version, it is very difficult to read and interpret individual graphs and their descriptions if only acronyms are given. The more so that the graphs (on the ordinate axis - y) also show only acronyms, so full understanding of the content of the figure requires searching for the full name of a given acronym in the text, which is a significant difficulty for the reader.

I think that in some part of the article, at the beginning or at the end, it is worth making a list of all the acronyms used in the study. There are so many acronyms that it is very difficult to read and analyze the content of the article, so it is worth listing all the acronyms in one place.

The article requires - in my opinion - minor language corrections.A large number of acronyms makes it difficult to read the text, so I suggest listing all acronyms together with their full name in one place.

Response 12: For the issue of abbreviating related terms (e.g. providing full names for Figure 7-10 figure names, etc.), the idea that the large number of acronyms makes it difficult to read the article is a view we share. For ease of understanding, and for the aesthetics of the graphics, the article includes a full explanation of the acronyms and full names in the Appendix to help the reader's understanding.

Point 13: I would like to ask about the error in the model analysis adopted by the authors. Did the authors take this error into account? I request information.

Response 13: Regarding the error in the model analysis used, the data in the results section takes into account losses in production, transport, processing and waste in the food-to-consumption process, but not in oil crops, which was considered at the writing stage of the article, but the addition of this error resulted in insignificant changes in the level of supply and demand for Uzbekistan. Also, the loss portion of oilseed crops is not directly no longer utilised, but is processed into feed, etc., which is also used indirectly by the population.

Point 14: Do the presented model studies allow to determine the prospects for balancing food supply and demand in Uzbekistan? How long a perspective can be developed based on the conducted research study? It is worth writing about it in the article.

Response 14: The proposed model study can determine the outlook for the balance of food supply and demand in Uzbekistan. The model can make forecasts by predicting changes in population size for a shorter number of years, as the energy for food and consumption changes, but this change cannot be better responded to by the model. Also, the main objective of this manuscript is to make recommendations on the relationship between food supply and demand and the achievement of self-sufficiency in Uzbekistan, without making forecasts on the outlook for supply and demand.

Point 15: Did the authors take into account in the analysis the nutritional models related to the changing preferences in terms of the quantity and quality of the components of the diet of the inhabitants of Uzbekistan? It's worth writing about.

Response 15: The manuscript takes into account the nutritional patterns associated with changes in the quantitative and qualitative aspects of the preferences of the Uzbekistan population's diet. A relevant analysis is carried out in the manuscript ‘3.2. Structural characteristics of food consumption’ on pp. 8-10.

We tried our best to improve the manuscript and made some changes marked in revised manuscript. We appreciate for reviewer’ s warm work earnestly and hope the correction will meet with approval. Once again, thank you very much for your comments and suggestions.

Round 2

Reviewer 2 Report

Thank you for introducing the changes and additions suggested in the review of the article, as well as all the clarifications regarding the detailed issues presented in the article.